# Changes in Physical and Chemical Properties of Thermally and Oxidatively Degraded Sunflower Oil and Palm Fat

**DOI:** 10.3390/foods9091273

**Published:** 2020-09-11

**Authors:** Berthold Wiege, Eberhard Fehling, Bertrand Matthäus, Marcus Schmidt

**Affiliations:** Max Rubner-Institut, Federal Research Institute of Nutrition and Food, Department of Safety and Quality of Cereals, 32756 Detmold, Germany; e.fehling@freenet.de (E.F.); bertrand.matthaeus@mri.bund.de (B.M.); marcus.schmidt@mri.bund.de (M.S.)

**Keywords:** deep-fat frying, sunflower oil, palm fat, dielectric constant, viscosity

## Abstract

Deep-fat frying is an important process used worldwide for the preparation of foods. Due to oxidation, hydrolysis, decomposition and oligomerization, numerous polar compounds are formed. These compounds change the physical, nutritional and sensory properties of the oil or fat. The standard methods of the German Society for Fat Science for the assessment of the quality of frying fats are time consuming and cost intensive. Therefore, alternative cost-effective and sensitive rapid methods, which ideally allow the quantitative determination of the quality of frying fats “in-line” in the deep-frying pan are needed. Sunflower oil and palm fat were thermally and oxidatively degraded in a beaker at atmospheric pressure under intensive stirring for 76 h at 175 °C. To evaluate the development of the physical properties during heat treatment, the viscosity and dielectric constant of these oils were measured. The temperature in a deep-frying pan can vary within a wide range (160–190 °C), and the viscosity and dielectric constant show a strong temperature dependence. Therefore, it was necessary to measure the temperature dependence of the viscosity and dielectric constant of the different degraded oils. Additionally, their chemical properties were characterized by high-performance gel permeation chromatography and Fourier-transform infrared spectroscopy (FTIR). The determination of the dielectric constant, which is directly correlated with the concentration of polar compounds, seems to be the best method for the assessment of the quality of used frying oils.

## 1. Introduction

Deep-fat frying is an important process used worldwide for the preparation of foods [1,2]. Under the conditions of high temperature (160–190 °C), air and moisture, a great number of chemical reactions occur in the oil. By oxidation, hydrolysis, decomposition and oligomerization, a large number of polar compounds evolve, which change the physical, physico-chemical, nutritional and sensory properties of the oil or fat [3,4]. Some routes of deterioration of frying fats and oils and methods for the evaluation of oil degradation have been described by several authors [5,6]. Frying oils that are highly deteriorated contain oxidized and polymerized compounds that might be harmful to human health [7,8,9]. Research has shown that the fraction of polar compounds isolated from oxidized oils is the most toxic to laboratory animals [10].

Usually, a huge amount of the frying oil is absorbed by the fried food. Thus, it is important to assure a high oil quality, using different parameters. These help to recognize the point at which the oil should be discarded to maintain optimum product quality [11]. In general, the quality of the product is directly correlated to the quality of the used frying medium.

The chemical parameters for the evaluation of oil quality are the amounts of polar and polymer compounds. However, the parameter recommended by the 3rd International Symposium on Deep-Fat Frying held in 2000 in Hagen/Germany for the analysis of suspect frying fats and oils is the total content of polar compounds [12]. The determination of the polar compounds is one of the most reliable methods for the assessment of used frying oils. Therefore, in several European countries, maximum values between 24% and 27% are set for commercially used frying oils.

One drawback of the determination of polar compounds is that the method is time-consuming and expensive. Therefore, the search for rapid test methods to determine the quality of frying oils during the processing is an ongoing challenge, and the 3rd International Symposium on Deep-Fat Frying recommended the use of rapid tests for monitoring oil quality [12]. These rapid tests should exhibit the following characteristics: (1) correlation with internationally recognized standard methods, (2) able to provide an objective index, (3) easy to use, (4) safe for use in a food processing/preparation area, (5) able to quantify oil degradation, (6) field rugged. Such rapid test methods include the measurement of the dielectric constant [13,14], which increases with increasing levels of polar material in the used frying oil, and the measurement of viscosity, which increases with increasing levels of polymers [15,16]. In 1986, Smith et al. [17] reported that the dielectric constant was the most convenient quality indicator in commercial deep-fat frying operation, and El-Shami et al. [18] found a proportional increase in the dielectric constant and viscosity with the heating time when the oil was heated between 180 and 190 °C. Wegmüller [13,14] found a good linear correlation between the dielectric constant measured by the food-oil sensor (Model NI-21A, Northern Instruments) and the concentration of Total Polar Compounds (TPC). Despite the good correlation of this rapid method with the standard method, there are some factors that affect the measurement. These include water, salt and minerals migrating from the food into the oil, but also, the type of oil has a strong influence on the result. Gertz [19] showed that the value of unheated coconut oil measured by the Food Oil Sensor (FOS) was 4.0 FOS units higher than the limit of 5.6 FOS units, corresponding to a content of 27% TPC.

The viscosity depends on the level of polymerization in the oil but also on the length of fatty acids and the oil type, respectively [19]. Suys [15] determined the viscosity of 116 samples of heated frying fats and 113 samples of heated frying oils and found that it was possible to estimate the content of polar compounds from the viscosity by calculating a second order equation. Kress-Rogers et al. [16] showed that the viscosity of frying media at 170 °C is correlated with the quality of the oil measured by the determination of polymerized and oxidized matter. Gertz [19] investigated 150 used frying oils with the Fri-Check^®^ instrument, which is based on the measurement of the viscosity in comparison to the TPM and polymerized triacylglycerols, and found strong correlation coefficients of 0.917 and 0.883, respectively. Additionally, Bansal et al. [20] found a good correlation between viscosity and polymer triacylglycerols.

Only the study of Kress-Rogers et al. [16] measured the changes in viscosity under frying conditions, while most of the other studies were performed with temperatures between 20 and 100 °C. Additionally, the measurement of the dielectric constant was mostly performed at lower temperatures.

Therefore, in this paper, the temperature dependence of the viscosity and dielectric constant of sunflower oil and palm fat during thermal and oxidative degradation was studied. Additionally, the heat-treated vegetable oils were characterized by high-performance gel permeation chromatography and FTIR-ATR spectroscopy.

## 2. Materials and Methods

### 2.1. Materials

Commercial refined sunflower oil and palm fat—a pure, unhardened vegetable fat—were purchased from a local German producer. The fatty acid composition is shown in Table 1.

### 2.2. Thermal and Oxidative Degradation of Oil Samples

In two open 1 L beakers, approximately 800 mL of sunflower oil and palm fat were heated continuously under intensive stirring at 175 ± 3 °C for 76 h. Test portions of about 100 mL were taken at different times and stored at 2–4 °C for further analysis.

The quality parameters of these two different types of vegetable oils subjected to heating at one temperature (175 °C)—which is the normal frying temperature—and for different times (*t* = 0, 12, 26, 40, 57 and 76 h) were characterized. The viscosity and dielectric constants of the twelve samples were then determined at different measuring temperatures. The measuring temperature for the dielectric constant was in the range of 20–130 °C, and that for the viscosity was in the range 20–170 °C.

### 2.3. Fatty Acid Analysis by Gas Chromatography (GC)

Based on the method of Tangkam et al. [21], the samples were prepared as follows: 20 mg of vegetable oil was dissolved in 1 mL of t-butyl-methylether. Next, 0.1 mL of the solution was thoroughly mixed with 0.05 mL of the methylation reagent trimethylsulfonium hydroxide (TMSH). Then, 5 μL was injected into the GC. The triacylglycerols react inside the injector with TMSH and produce fatty acid methyl esters.

The samples were analyzed on a gas chromatograph (HP 5890, Hewlett Packard, Waldbronn, Germany) using a HP-FFAP column (25 m × 0.2 mm). The temperatures of the injector and detector were set to 280 °C, and nitrogen was used as a carrier gas. The initial column temperature was set to 70 °C for 1 min. Then, it was raised to 150 °C at 10 °C/min and to 240 °C at 5 °C/min, and maintained at this final temperature for 20 min at 240 °C.

### 2.4. Viscosity

The viscosity of the sunflower oil and palm fat in the liquid state was measured using the viscotester VT 550 with the measuring equipment MV 1 (Haake Meßtechnik GmbH & Co, Karlsruhe, Germany). This rotatory viscometer allows the determination of the viscosity at shear rates in the range of 1 to 1872 s^−1^. A circulating thermostat (F 25-MV, Julabo) filled with silicone oil was used to keep the temperature constant in the range of 20 to 170 °C. At 170 °C, the accuracy of the temperature was approximately ±2 °C, whereas at lower temperatures, the accuracy was considerably better.

### 2.5. Dielectric Constant

For the measurements of the dielectric constant ε and its dependence on temperature, the Dipolmeter DM 01 from Wissenschaftlich—Technische Werkstätten GmbH (Weilheim, Germany) in combination with a F 25-MV thermostat (Julabo, Seelbach, Germany) containing silicone oil was used. The measuring cell for the liquids (DFL 1) was filled with 20 mL of the vegetable oil. The dielectric constant was determined at a frequency of approximately 2.0 MHz. The measuring range of DFL 1 is ε = 1–3.6, and the relative accuracy is Δε/ε = 4 × 10^−5^. The cell temperature was calibrated in relation to the temperature of the silicone oil bath. Cyclohexane (99.5%), carbon tetrachloride (99.5%), o-xylole (>99%) and propionic acid (99.5%) (at 10 and 40 °C) were employed for the calibration of the dielectric constant in the range of ε = 2.023–3.440.

### 2.6. High-Performance Gel Permeation Chromatography (HPGPC)

The HPGPC system consists of a Kontron Instruments pump 422 (Milan, Italy) equipped with a medium type pump head, an ERC 7515A refractive index (RI) detector (ERC Inc., Kawaguchi-City, Japan) with a temperature setting of the optical block of 35 °C, a Rheodyne 7725i sample injector (Cotati, CA, USA) equipped with a 100 μL sample loop, a Jetstream 2 column oven set to 30 °C (Thermotechnic Products, Langenzersdorf, Austria) and a Kromasystem 2000 data system (Kontron Instruments). Separations were carried out on a set of three columns of different porosity connected in series (PLgel 5 μm: 100 Å, 1000 Å, 10,000 Å, each 300 × 7.5 mm, with a precolumn of 50 × 7.5 mm; Polymer Laboratories, Shropshire, UK) so that a linear separation range of about 100 to >500,000 g/mol was achieved. The samples were dissolved in tetrahydrofurane (THF) stabilized with 0.025% butylhydroxytoluene (BHT) for GPC (Fisher Scientific, Loughborough, UK) at a concentration of 3% (*w*/*v*), and the flow rate of the THF was set at 1 mL/min. Polystyrene standards (Polymer Standards Service, Mainz, Germany) were used for calibration.

### 2.7. Fourier-Transform Infrared Spectroscopy (FTIR)

The FTIR spectroscopy of samples was carried out using the FTIR spectrometer IFS 28 from Bruker-Franzen Analytik GmbH (Bremen, Germany). The optical system included a permanent adjusted interferometer, a potassium bromide substrate beam splitter and a high resolution DLATGS detector. For the determination of the spectra, 30 scans at a resolution of 1 cm^−1^ were collected in the range of 4000–650 cm^−1^. A horizontal ATR unit (A537) with a 45° ZnSe crystal was used for the measurements. In the case of the solid palm fat samples, it was necessary to melt 0.3–0.4 g of the fat at a temperature of 50–60 °C on the ATR crystal and to measure the spectra in the liquid state.

## 3. Results

### 3.1. Viscosity

For the investigations, palm fat and sunflower oil were used, which are widespread in the preparation of food. Table 1 presents the fatty acid compositions of both oils, showing that the main difference is the remarkably higher content of saturated fatty acids in palm fat (about 56%) and the markedly higher content of mono- and polyunsaturated fatty acids in sunflower oil (about 86%). Based on these differences, a significantly higher thermal stability of palm fat compared to that of sunflower oil can be expected.

The increase in viscosity during the thermal treatment of vegetable oils results from the formation of oligomer molecules. The chemistry of the uncatalyzed and catalyzed thermal polymerization of vegetable oils, their fatty acids and methyl ethers is well described in the literature [22,23,24]. Heat polymerization, which results from the presence of olefinic double bonds [22], can be studied by observing the increase in viscosity with reaction time [25].

During the thermal and oxidative treatment of vegetable oils, the polymerization of unsaturated triacylglycerols can also be caused by the formation of ether bridges. The dependence of the dynamic viscosity of degraded sunflower oil and palm fat samples on the measuring temperature and the time of thermo-oxidative treatment is presented in Figure 1.

A two-parameter fit according to the Arrhenius–Andrade equation (ln (η) = ln (A) + B/T) [26] is not suitable for fitting the measured values of viscosity against temperature.

However, the dependence of the viscosity (η) on the temperature, especially across a wide temperature range, can be excellently described by the formula $\ln\eta =A+\;\frac{B}{T}+\frac{C}{T^{2}}\;$, with the empirical constants A, B and C and the temperature T given in Kelvin [27].

The functions ln η = f(1/T) for the different degraded sunflower oils and palm fats are presented in Figure 2.

The corresponding fit parameters A, B and C are shown in Table 2.

At a constant measuring temperature, a strong increase in the viscosity with the time of thermo-oxidative treatment was observed. Sunflower oil clearly shows a stronger increase in viscosity in comparison to palm fat (Figure 3).

This behavior can be explained by the higher concentration of polyunsaturated fatty acids in sunflower oil in comparison to that in palm fat. The viscosity of treated and untreated sunflower oil does not depend on the shear rate in the range of 234–1872 s^−1^ at all measuring temperatures (Table 3). Sunflower oil and palm fat (in the liquid state) can be regarded as Newtonian fluids.

### 3.2. Dielectric Constant

The dielectric constant (ε) of vegetable oils and fatty acid methyl esters is directly related to the structure of these compounds [28]. In the homologous series of fatty acid methyl esters, the dielectric constant, which was measured at constant temperature and constant frequency, decreases significantly with increasing length of the fatty acid chain [28]. This behavior can be described quantitatively by the Smittenberg relation [29]. In the series of C18 carbonic acid methyl esters C18:0 to C18:3, the dielectric constant at 40 °C increases from 3.021 for stearic acid up to 3.349 for linolenic acid methyl ester. With an increasing number of double bonds, the electronic polarizability becomes larger, and therefore, ε rises.

During frying or the heating of vegetable oils under atmospheric conditions, various chemical processes such as oxidation, hydrolysis, polymerization and cleavage take place [30]. The reaction with oxygen creates hydroperoxides, epoxides, alcohols, ketones, fatty acids and other minor components. Furthermore, oligomeric triacylglycerols emerge by the intermolecular formation of ether bridges and C-C bonds. However, the reaction with oxygen causes a strong increase in polar compounds in the vegetable oil during heating or frying, which causes an increase in the dielectric constant with time [31].

The dependence of the dielectric constant of sunflower oil and palm fat on the time of thermo-oxidative treatment is presented in Figure 4.

The dielectric constants of the treated and untreated sunflower oil and palm fat are nearly identical at both measuring temperatures. At a measuring temperature of 130 °C, the dielectric constant increases by approximately 23% over 76 h. Since the dielectric constant of sunflower oil increases visibly faster over time, compared to that of palm fat, a lower thermal stability is evident.

The increase in polar compounds is combined with an increase in the mean dipole moment μ of the oil phase. A relation of ε, μ and *T* is given by Equation (1), of Debye, Clausius and Mosotti [32]:(1)$$P_{m}=\;\frac{\varepsilon \;-\;1}{\varepsilon \;+\;2}*\frac{M}{\rho }\;=\;\frac{4}{3}*\pi *\;N_{A}*\;\left(\alpha_{e}+\;\alpha_{a}+\;\frac{{µ}^{2}}{3kT}\right)$$
$P_{m}$ = molecular polarization, $\alpha_{a}$ = atomic polarizability, m = mean molecular mass, ε = dielectric constant, *N_A_* = Avogadro’s number, *ρ* = density, μ = dipole moment of the molecule, *k* = Boltzman constant, $\alpha_{e}$ = electronic polarizability, *T* = temperature.

The temperature dependence of the dielectric constant ε presented in Figure 5 can be explained by the term (μ^2^/3 kT) of Equation (1).

The orientational polarization of the molecules (μ^2^ / 3 kT) decreases with increasing temperature. Since the density and the atomic and electronic polarizability are nearly independent of temperature, the temperature dependence of ε in Figure 5 can be explained by the temperature dependence of the orientational polarization. In addition, this confirms the results of Pecovska-Gjorgjevich [33], who demonstrated a decrease in DK with increasing temperature as well. The comparison of the absolute DK values of the two samples also provides evidence for the better heat stability of palm fat compared to that of sunflower oil. While after 0 h treatment, the DK curves are nearly identical, after 57 or 76 h of treatment, the sunflower oil shows higher DK values.

### 3.3. High-Performance Gel Permeation Chromatography (HPGPC)

HPGPC is considered to be a very efficient method for the analysis of thermally degraded vegetable oils. Hansen et al. [34] studied the polymerization of triolein, which was heated at 190 °C for 60 h. They observed the formation of dimeric, trimeric and tetrameric triacylglycerols. During heating or frying, a wide variety of chemical reactions result in the formation of compounds with high molecular weight and polarity [35]. In this study, potatoes were fried in sunflower oil for 8 min at an initial temperature of 180 °C. After a total of sixty frying sets, an increase in polar compounds from 3.8% in the unused oil up to 27.3% in the used sunflower oil was observed. The triacylglycerol polymers and dimers, which were determined by HPSEC, increased from basal values of 0.06% and 0.50% up to 5.39% and 10.91% (*w*/*w*) in the oil after sixty frying steps [35].

In the high-performance gel permeation chromatograms of the thermo-oxidatively treated sunflower oil and palm fat, four peaks were observed. By the calibration of the molecular weight of the products with polystyrene standards, these peaks were identified as triacylglycerol monomers, dimers, trimers and tetramers. Higher oligomeric triacylglycerols did not occur. The change in the concentration of oligomeric triacylglycerols in the sunflower and palm fats is shown in Figure 6.

Because of the lower thermal stability of sunflower oil in comparison to that of palm fat, the decrease in the monomeric triacylglycerols in sunflower oil was significantly stronger. After 76 h of thermo-oxidative treatment, only 38% and 54% of the original existing triacylglycerol monomers were observed in sunflower oil and palm fat, respectively.

### 3.4. Fourier-Transform Infrared Spectroscopy (FTIR)

FTIR spectroscopy is a useful method for the quality control of vegetable oils and fats. The carbonyl groups of the triacylglycerols give rise to a very intense band at 1742 cm^−1^, whereas the alkyl chains show a number of bands in the so-called fingerprint region (1450–900 cm^−1^). The absorption band at 967 cm^−1^ is characteristic for isolated trans double bonds. Cis double bonds are characterized by their CH stretching band at approximately 3009 cm^−1^. The absorption bands corresponding to the cis double bonds were integrated in the region of 2988–3029 cm^−1^ for sunflower oil and 2991–3021 cm^−1^ for palm fat. The results of the integration are presented in Table 4.

The integrals of the absorption bands with peak maxima of 3008.4 and 3005.6 cm^−1^ for sunflower oil and palm fat, respectively, decrease significantly with increasing time of thermo-oxidative degradation.

## 4. Discussion

During deep-fat frying, the oil or fat is degraded by oxidation, hydrolysis, decomposition and oligomerization, and therefore, the physical, nutritional and sensory properties are changed with increasing time of frying. Since a large quantity of the frying fat is absorbed by the fried goods, effective quality control for the frying fat is necessary. The chemical standard methods for the determination of the quality of used frying fats are very time consuming and expensive. Therefore, the 3rd International Symposium on Deep-Fat Frying recommended the use of rapid tests for monitoring oil quality [12].

The dielectric constant increases with increasing time of thermal treatment, if monitored at constant measuring temperatures (see Figure 4). However, a comparison of the thermal–oxidative oligomerization with the increase in dielectric constant is difficult. The problem is that the oligomerization of the triglycerides occurs not only by the formation of ether bridges. The oligomerzation can also take place via C-C bonds between the triglycerides. However, in this case, the polarity and therefore the dielectric constant are not influenced by the oligomerization process. The decrease in ε with increasing temperature (see Figure 5) is, within the relatively small measuring range, nearly linear, with the exception of the 0 and 12 h isobars. However, in principle, across a wider temperature range, the decrease in ε with increasing temperature cannot be linear. Since at high temperature, the term μ^2^/ 3 kT tends to zero, ε is, according to formula (1), only dependent on the electronic and atomic polarizability, which are nearly independent of temperature. Since the atomic polarizability is only about 10% of the electronic polarizability, the molecular polarization P_m_ tends to pass into the molar refraction R_m_. That means ε turns into n^2^ (the square of the refractive index n). Since the refractive index of palm fat and sunflower oil is about 1.45–1.47 [36], in both cases, ε tends to ε = 1.46^2^ = 2.13 (compare this value with the 0 h isobars in Figure 5). That means an asymptotic approximation of ε from 2.75 (both oils, Figure 5—0 h, 130 °C) to 2.13 × 1.1 = 2.34 at very high temperatures—if one also considers the influence of the atomic polarizability.

## 5. Conclusions

The dielectric constant ε, which is well correlated with the concentration of polar compounds in the frying fat, allows a quantitative determination of the quality. Since the temperature in the fryer can vary in the range of approximately 160–190 °C and ε shows a strong temperature dependence, it was necessary to determine the temperature dependence of ε for different deteriorated frying fats. The experimental results of ε = f(T,*t*) presented in Figure 4 and Figure 5 can be the basis of a measuring instrument for the determination of the quality of frying fats in-line, e.g., in industrial fryers. The measuring instrument has to determine both the temperature and the dielectric constant of the frying fat. By the inclusion of a mathematical model, it is then possible to calculate ε at a well-def ined standard temperature and to determine the concentration of polar compounds in the frying fat. The amount of polar compounds can then be compared with the recommendations of the German Society for Fat Science for used frying fats and allow the determination of the period after which the fat has to be changed.

## Figures and Tables

**Figure 1 foods-09-01273-f001:**
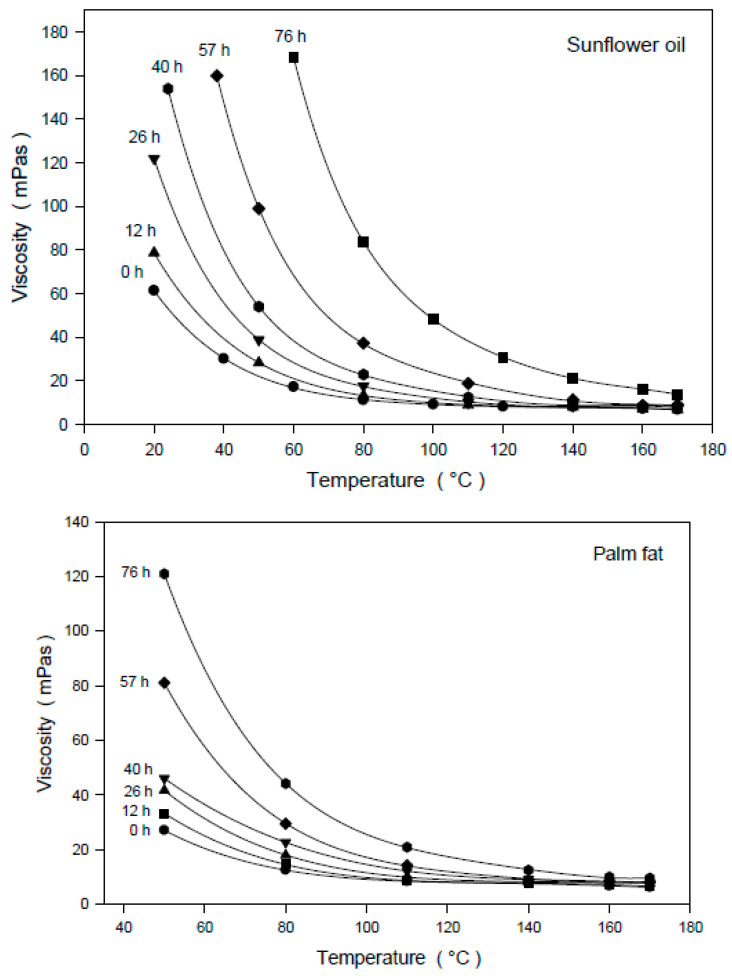
Dependence of dynamic viscosity of thermo-oxidatively treated sunflower oil and palm fat on the measuring temperature; treatment: 0–76 h at 175 ± 3 °C; normal atmosphere; shear rate: 1872 s^−1^; parameter: thermal treatment time.

**Figure 2 foods-09-01273-f002:**
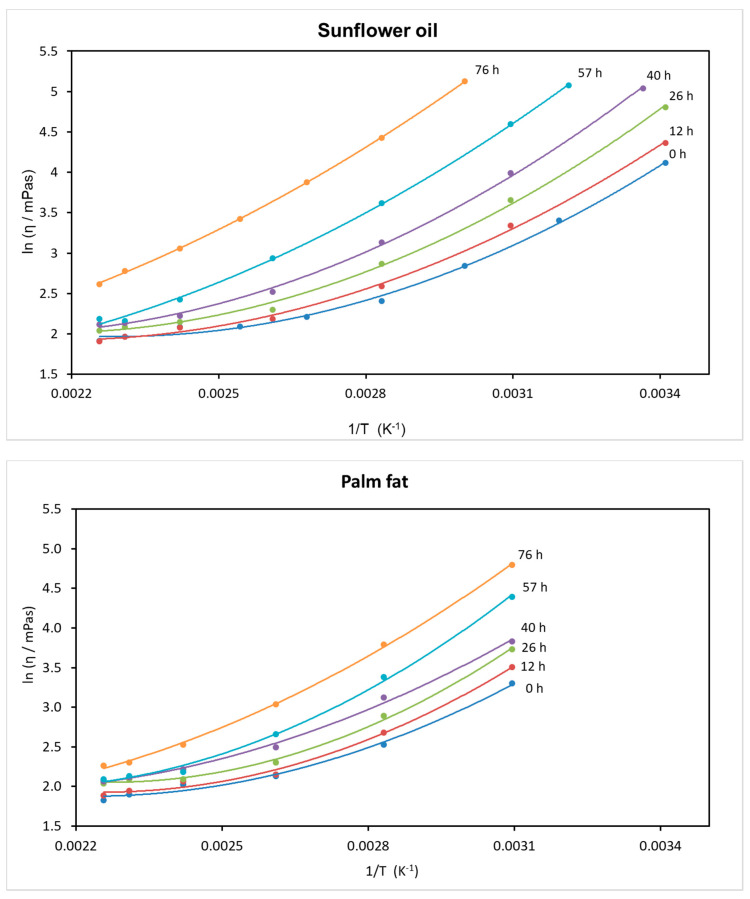
The dependence of viscosity η on the temperature of the sunflower oil and palm fat; ln η = f(1/T), approximated by a three-parameter fit.

**Figure 3 foods-09-01273-f003:**
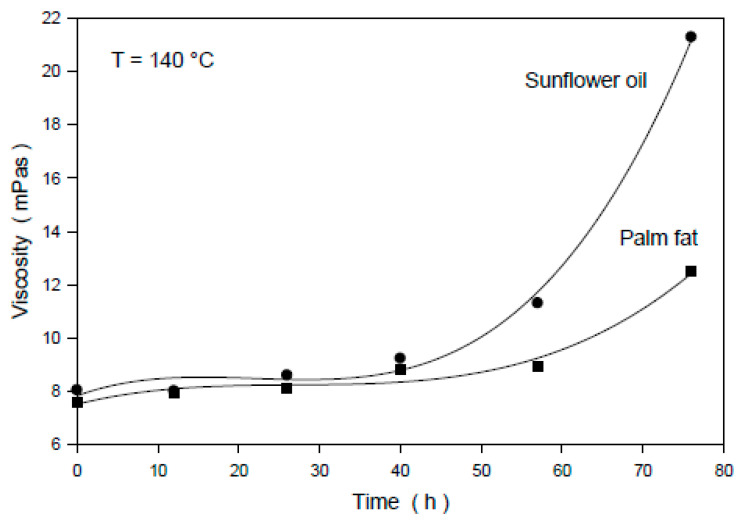
Dependence of the dynamic viscosity of sunflower oil and palm fat on the time of thermo-oxidative treatment; measuring temperature: 140 °C; shear rate: 1872 s^−1^.

**Figure 4 foods-09-01273-f004:**
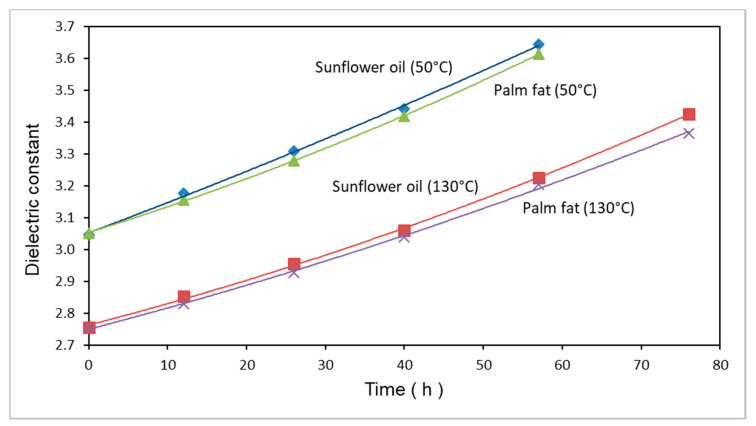
Dielectric constant as a function of the time of the thermo-oxidative degradation of sunflower oil and palm fat at two measuring temperatures; thermal degradation: 0–76 h at 175 ± 3 °C.

**Figure 5 foods-09-01273-f005:**
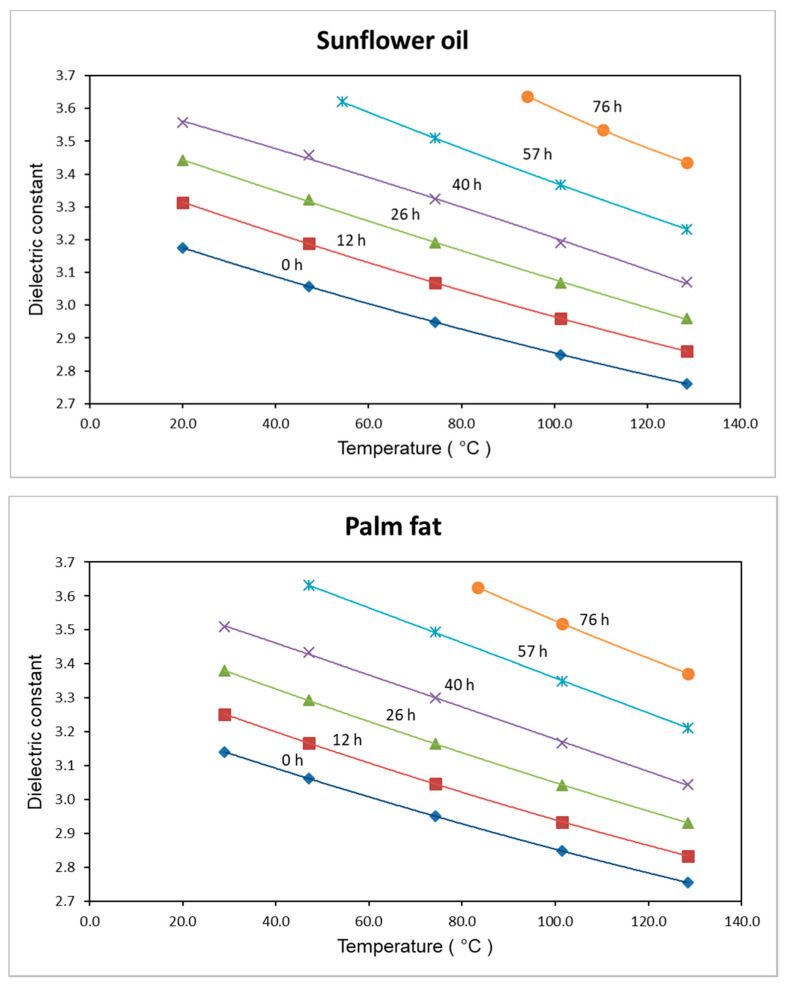
Dielectric constants of thermo-oxidatively treated sunflower oil and palm fat vs. measuring temperature; temperature of thermal treatment: 175 ± 3 °C; parameter: thermal treatment time.

**Figure 6 foods-09-01273-f006:**
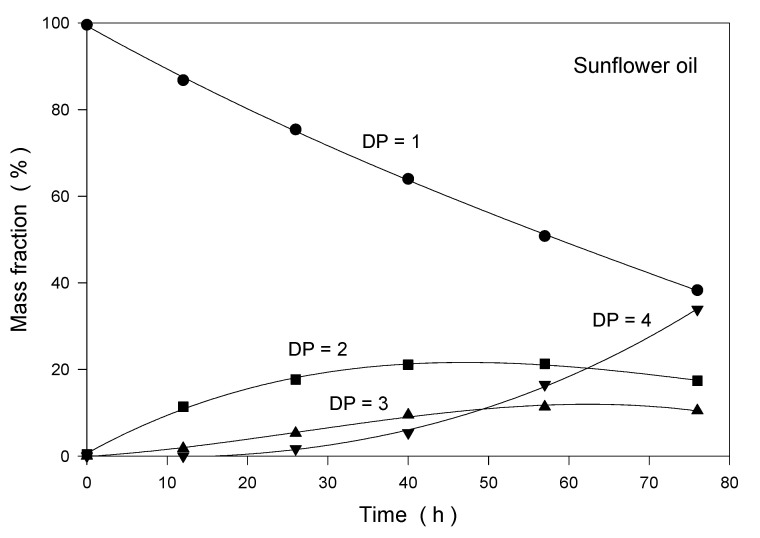
Kinetics of the oligomerization of sunflower oil and palm fat at 175 °C under normal atmosphere; DP = degree of polymerization.

**Table 1 foods-09-01273-t001:** Fatty acid composition of sunflower oil and palm fat.

Fatty Acid	Sunflower Oil (%)	Palm Fat (%)
C12:0	-	0.6
C14:0	-	1.5
C16:0	7.1	48.2
C18:0	4.2	5.4
C18:1	20.5	34.3
C18:2	65.6	8.9

**Table 2 foods-09-01273-t002:** Mathematical description of the viscosity η as a function of temperature. Fit parameters of the function $\ln\eta =A+\;\frac{B}{T}+\frac{C}{T^{2}}$ for sunflower oil and palm fat.

**Sunflower Oil**				
***t* (h)**	**A**	**B (K)**	**C (10^6^ × K^2^)**	**R^2^**
0	10.938	−7841.6	1.7135	0.9965
12	9.455	−6938.7	1.5982	0.9979
26	10.051	−7507.4	1.7523	0.9979
40	9.113	−7007.9	1.7249	0.9988
57	4.992	−4344.5	1.3612	0.9990
76	3.578	−3247.9	1.2535	0.9999
**Palm Fat**				
***t* (h)**	**A**	**B (K)**	**C (10^6^ × K^2^)**	**R^2^**
0	11.026	−8232.9	1.8518	0.9924
12	13.186	−9997.1	2.2193	0.9942
26	14.757	−11,210.1	2.4727	0.9973
40	8.118	−6209.1	1.5611	0.9965
57	11.781	−9502.1	2.3017	0.9974
76	6.398	−5442.6	1.5928	0.9993

**Table 3 foods-09-01273-t003:** Dependence of the dynamic viscosity η (mPa x s) of sunflower oil on the shear rate and time of thermo-oxidative treatment; measuring temperature T = 80 °C.

Shear Rate (s^−1^)	0 h	12 h	26 h	40 h	57 h	76 h
234	9.5	11.2	17.3	22.4	36.4	83.7
468	10.0	12.0	17.1	22.3	37.0	83.9
702	10.3	12.6	17.2	22.4	37.0	83.8
1403	10.8	13.1	17.5	22.8	37.3	83.7
1872	11.1	13.3	17.6	22.9	37.3	83.7

**Table 4 foods-09-01273-t004:** Integral of the infrared absorption band of cis double bonds of the fatty acids of sunflower oil and palm fat, with dependence on the time of thermo-oxidative degradation.

Time (h)	Sunflower Oil Int. (3008.4 cm^−1^)	Palm Fat Int. (3005.6 cm^−1^)
0	2.010	0.485
12	1.853	0.407
26	1.691	0.335
40	1.521	0.267
57	1.288	0.178
76	1.019	0.109

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
