# Peer review of "Changes in Physical and Chemical Properties of Thermally and Oxidatively Degraded Sunflower Oil and Palm Fat"

_foods, 2020, doi:10.3390/foods9091273_

Round 1

Reviewer 1 Report

Authors are asked to make the changes to the paper according to the comments below. A deeo review of the discussion, which appears too brief, is strongly recommended. overall, the work does not seem to bring significant elements of novelty and originality to the scientific community of reference.

lines 11-12
please, authors check the sentence "...a large number of polar compounds
evolve which change the physical, nutritional and sensory properties of the oil or fat".

line 14
please, authors verify to change the word "inexpensive".
Probably it will impossible to find an absolutely inexpensive method.

Line 16
please, authors check the term "online". do they intend "in real time"?

lines 38-41
Please authors shorten this sentence, and rearrange it in a more clear way

Line 43
Please, authors describe shortly also the other chemical parameters generally used for the evaluation of degradation in food oil quality-

line 53
please, authors consider to put a dot before the list of the characteristics, so before "(1) correlate...)

LINE 64
please, authors check the sentence "...some factors which affecting the measurement.."

line 94
Please, authors evaluate to change "continuous" with "continuously"

Line 96
Please, authors add a reference for the method of FA analysis by GC

lines 161-163
Please, author check the label on the x-axis of the graphs. i can read temperature on the x-axis, but i can view only strange characters for the measure unit (i don't read °C)
Furthermore, i can see 2 graphs for sunflower oil, reporting viscosity vs. temperature, and the 2 graphs seem to be equal. then there is a third graph for the palm oil.

line 172
please, author check the character in the graph close to the word temperature, i can't see "°C" but a strange character. maybe there is a problem with some characters in the pre-print of the pdf file, only for graphs.

line 200
please authors refer to the previous comments in line 172

lines 218-219
please authors refer to the comments reported in the line 161-163

Author Response

All reviewer comments have been answered below in a point-by-point manner.  We wish to take this opportunity to thank the reviewer for the many very helpful comments. The authors are confident that we have addressed all of the issues raised in this review process and believe that the new version of our manuscript is much improved.

Authors are asked to make the changes to the paper according to the comments below. A deeo review of the discussion, which appears too brief, is strongly recommended. overall, the work does not seem to bring significant elements of novelty and originality to the scientific community of reference.

As suggested, the discussion was re-written to discuss the results of the research in much more detail and compare them with the relevant existing research.

To the best of the authors knowledge, this is the first research work investigating the suitability of the dielectric constant for the real time analysis of quality deterioration of frying oils. In addition, quality deterioration as a result of heat induced polymerization can be determined by an increase in apparent viscosity. However, the authors revised the manuscript to make the novelty of the approach clearer to the reader.

lines 11-12 please, authors check the sentence "...a large number of polar compounds
evolve which change the physical, nutritional and sensory properties of the oil or fat".

The sentence was re-written and split into two. It reads now as follows: “Due to oxidation, hydrolysis, decomposition and oligomerization numerous polar compounds are formed. These compounds change the physical, nutritional and sensory properties of the oil or fat.

line 14 please, authors verify to change the word "inexpensive".
Probably it will impossible to find an absolutely inexpensive method.

As suggested, the word “inexpensive” was changed to “cost-effective”.

Line 16 please, authors check the term "online". do they intend "in real time"?

Here and throughout the manuscript, the term “online” was changed to “in-line”, which is defined by the Oxford dictionary as „Taking place or situated as an integral part of a continuous, usually linear, sequence of operations or machines (as in an assembly line)“.

lines 38-41 Please authors shorten this sentence, and rearrange it in a more clear way

As suggested, the sentence was shortened by dividing it into 3 and revised for better understanding. It reads now as follows: “Usually, a huge amount of the frying oil is absorbed by the fried food. Thus, making it important to assure a high oil quality, using different parameters. These help to recognize the point at which the oil should be discarded to maintain optimum product quality [11].”

Line 43 Please, authors describe shortly also the other chemical parameters generally used for the evaluation of degradation in food oil quality-

As suggested, the authors shortly introduce the other chemical parameter for the evaluation of frying oil and fat, namely the total polymer content.

line 53 please, authors consider to put a dot before the list of the characteristics, so before "(1) correlate...)

A double point (:) was included before the list of the characteristics.

LINE 64 please, authors check the sentence "...some factors which affecting the measurement.."

The sentence was revised and reads now as follows: “…some factors, which affect the measurement…”

line 94 Please, authors evaluate to change "continuous" with "continuously"

As suggested, the term “continuous” was changed to “continuously”.

Line 96 Please, authors add a reference for the method of FA analysis by GC

A reference for the FA analysis by GC was added.

lines 161-163 Please, author check the label on the x-axis of the graphs. i can read temperature on the x-axis, but i can view only strange characters for the measure unit (i don't read °C)
Furthermore, i can see 2 graphs for sunflower oil, reporting viscosity vs. temperature, and the 2 graphs seem to be equal. then there is a third graph for the palm oil.

The label on the x-Axis is not displayed correctly in the pdf-file. In the original word-file submitted it is displayed correctly as °C. the authors will take care that this error is not repeated upon resubmission of the manuscript.

The appearance of 2 graphs for sunflower oil in Figure 1 is an error that has been corrected. Now Figure 1 only contains 1 diagram for sunflower oil and another one for palm fat.

line 172 please, author check the character in the graph close to the word temperature, i can't see "°C" but a strange character. maybe there is a problem with some characters in the pre-print of the pdf file, only for graphs.

Like for the previous comment, the strange character in place of °C originates from an error in the conversion of the word-file into pdf format. The authors will take care that the axis labelling is displayed correctly when re-submitting the manuscript.

line 200 please authors refer to the previous comments in line 172

The same response as for the previous comment in line 172 applies.

lines 218-219 please authors refer to the comments reported in the line 161-163

The same response as for the previous comment in line 161-163 applies.

Author Response

All reviewer comments have been answered below in a point-by-point manner. We wish to take this opportunity to thank the reviewer for the many very helpful comments. The authors are confident that we have addressed all of the issues raised in this review process and believe that the new version of our manuscript is much improved.

Some of the replies of the authors were in this or in a modified form included into the manuscript.

The authors have characterized the quality parameters of two different types of vegetable oils subjected to heating at different temperatures and for different times.

The authors have characterized the quality parameters of two different types of vegetable oils subjected to heating at only one temperature (175 °C) - which is the normal frying temperature - and for different times (t = 0, 12, 26, 40, 57, 76 h). Dielectric constant and viscosity of these twelve samples were then determined at different measuring temperatures. Measuring temperature of the dielectric constant was in the range of 20-130°C and of viscosity in the range 20-170°C (s. Fig 4 and Fig. 1)

Viscosity was found to increase with temperature and time of exposure as a result of oxidation of oil with this increase being more pronounced for unsaturated sunflower oil.

Viscosity was found to increase with increasing time of thermal treatment (see Fig. 2 ; t = 0, 12, 26, 40, 57, 76 h) as a result of oxidation of oil. With increasing measuring temperature the viscosity of each of the twelve samples decreases (see Fig. 1).

The dielectric constant was found to decrease with temperature and treatment time.

The dielectric constant was found to increase with time of thermal treatment if compared at the same measuring temperature (s. Fig. 4). Because of the formation of ether bridges (polymerisation of triglycerides caused by oxygen) the polarity of the reaction products increases and therefore also the dielectric constant increases with the time of thermal treatment. The formation of ether bridges causes an increase of the electronic polarizability αe and therefore according to formula 1 (line 227) an increase of the dielectric constant with increasing time of thermal treatment.

  1. It was suggested that the authors evaluate the constants A and B in the equation 
  2. Thanks a lot for this suggestion. We have done it and found that a quadratic fit is much better suited to describe the data ln η= A + B/T + C/T2. The plots for sunflower oil and palm fat are shown in the manuscript together with the 3 fit parameters of each of the 12 samples.
  3. ln η= A + B/T and include these either in a table or fit them as a function of time. This will be more meaningful in correlation the degradation with time and temperature.

  1. It is also suggested that the authors correlate the decrease of dielectric constant with time and temperature with composition (DP1, DP2, DP3 and DP4) and include this correlation for both oils.
  2.  

The dielectric constant increases with increasing time of thermal treatment – if compared at constant measuring temperatures (s. Fig. 3). However, the suggestion to compare the thermal (oxidative) oligomerization with the increase of dielectric constant is very meaningful. The problem is, that the oligomerization of the triglycerides happens not only by formation of ether bridges.

An oligomerzation can also occur via C-C bonds between two triglycerides. But in this case the polarity and therefore the dielectric constant are not influenced by the oligomerisation process.

Minor comments

  1. Fig. 1 has three panels. It is not clear whether the first and second panels are identical.Yes, you are right. It was a computer error. It´s corrected.
  2.  
  3.  
  4. Figs. 2-4, delete the symbols that appear next to temperature value and replace with C?It was a computer error. It´s corrected.
  5.  
  6.  
  7. Include error bars in all figures.This is not possible! Only single determinations were carried out. However, because of the self-consistency of the experimental values and the high quality of the measuring results the errors bars would be very small.
  8.  
  9.  
  10. Line 225 – Is the decrease in μ2/ 3 kT with temperature linear? 
  11. Is it consistent with observed linear decrease of ε with temperature?

The first derivative of μ2/ 3 kT according to the temperature is:

d (μ2/ 3 kT) / d (T) = - μ2/ 3 kT2

This is a nonlinear decrease with increasing temperature.

Is it consistent with observed linear decrease in ε with temperature.

The decrease of ε with increasing temperature is within the relatively small measuring range nearly linear with exception of the 0 h and 12 h curve. However in principal in a wider temperature range the decrease of ε with increasing temperature can´t be linear. Since at high temperature the term μ2/ 3 kT tends to zero, ε is according to formula 1 (line 227) only dependend on the electronic and atomic polarizability, which are nearly independent from temperature. Since the atomic polarizability is only about 10% of the electronic polarizability, the molecular polarization Pm tends to pass into the molar refraction Rm. That means ε turns into n2 (the square of the refractive index n).

Since the refractive index of palm fat and sunflower oil is about 1.45 – 1.47 (http://www.dgfett.de/material/physikalische_eigenschaften.pdf)

in both cases ε tends to ε = 1.46 2 = 2.13 (Fig. 4). That means an asymptotic approximation of ε from 2.75 (both oils Fig. 4 – 0h, 130°C) to 2.13 x 1.1 = 2.34 at very high temperatures - if you also consider the influence of the atomic polarizability.

Round 2

Reviewer 2 Report

Authors have addressed the reviewer's comments satisfactorily. The manuscript is acceptable in its present form.